# A 13-Gene DNA Methylation Analysis Using Oral Brushing Specimens as an Indicator of Oral Cancer Risk: A Descriptive Case Report

**DOI:** 10.3390/diagnostics12020284

**Published:** 2022-01-23

**Authors:** Roberto Rossi, Davide B. Gissi, Andrea Gabusi, Viscardo Paolo Fabbri, Tiziana Balbi, Achille Tarsitano, Luca Morandi

**Affiliations:** 1Section of Oral Sciences, Department of Biomedical and Neuromotor Sciences, University of Bologna, 40125 Bologna, Italy; roberto.rossi38@unibo.it (R.R.); andrea.gabusi3@unibo.it (A.G.); 2Department of Biomedical and Neuromotor Sciences, Section of Anatomic Pathology “M. Malpighi”, Bellaria Hospital, 40125 Bologna, Italy; viscardopaolo.fabbr2@unibo.it; 3Unit of Anatomic Pathology S. Orsola Hospital, IRCCS Azienda Ospedaliero Universitaria, 40138 Bologna, Italy; tiziana.balbi@aosp.bo.it; 4Maxillo-Facial Surgery Unit, Department of Biomedical and Neuromotor Sciences, IRCCS Azienda Ospedaliero Universitaria Bologna, University of Bologna, 40138 Bologna, Italy; achille.tarsitano2@unibo.it; 5Functional and Molecular Neuroimaging Unit, Bellaria Hospital, Department of Biomedical and Neuromotor Sciences, University of Bologna, 40139 Bologna, Italy; luca.morandi2@unibo.it; 6Functional and Molecular Neuroimaging Unit, IRCCS Istituto delle Scienze Neurologiche di Bologna, 40139 Bologna, Italy

**Keywords:** oral leukoplakia, oral squamous cell carcinoma, diagnosis, prognosis, brushing, DNA methylation

## Abstract

Analysis of genetic or epigenetic markers from saliva or brushing specimens has been proposed as a diagnostic aid to identify patients at risk of developing oral cancer. However, no reliable non-invasive molecular method for this purpose is commercially available. In the present report, we describe the potential application of a procedure based on a 13-gene DNA methylation analysis using oral brushing samples from a patient affected by oral leukoplakia who developed two metachronous oral carcinomas during the follow-up period. A positive or a negative score was calculated for each brushing sample based on a predefined cut-off value. In this patient, a positive score was detected in the oral leukoplakia diagnosed more than 2 years before the development of oral squamous cell carcinoma and subsequently in clinically healthy mucosa 8 months before the appearance of a secondary tumor. This suggests a potential role of our procedure as an indicator of oral cancer risk.

## 1. Introduction

Oral squamous cell carcinoma (OSCC) is the most frequent neoplastic disease of the oral cavity. It usually affects patients in the sixth to eight decades of life, despite recently there is an increasing incidence of OSCC in younger individuals. Elderly OSCC patients generally report usual exposure to well-known risk factors for cancer insurgence, such as tobacco or alcohol consumption [1]. The tongue is the most common site of OSCC insurgence despite the age of presentation, even though some authors reported the alveolar process as a predilection OSCC site in elderly patients [2]. OSCC is commonly preceded by oral potentially malignant disorders (OPMDs), of which oral leukoplakia (OL) is the most common [3]. In addition, patients affected with OSCC can develop loco-regional recurrence during aftercare at a rate of 40% [4], and recent studies reported higher recurrence percentages in younger patients compared to elderly ones [1]. The clinical and histological features of a single OPMD do not provide sufficient information for identifying patients who will develop OSCC during follow-up. For example, the presence and grade of epithelial dysplasia is the most reliable parameter when assessing the risk of OPMD malignant transformation, and the more severe the degree of dysplasia, the greater the likelihood of progression to malignancy. Nevertheless, not all OPMDs with dysplasia undergo neoplastic transformation, as malignancies have been observed to arise from non-dysplastic OPMD as well [5]. Conventional oral examinations and incisional biopsy with histological assessment are used to identify and follow up patients at risk of OSCC insurgence [6]. However, early diagnosis of OSCC can be hindered by the practitioners’ clinical experience, and the subtle nature of the lesions may be undetectable by oral examination alone. Incisional biopsy is an invasive surgical approach that creates discomfort and is not applicable as a routine diagnostic tool during follow-up of high-risk OSCC patients [7]. Thus, the development and application of non-invasive diagnostic tools are crucial. Several studies have suggested the use of biomarkers for early detection and monitoring of OSCC patients using non-invasive or minimally invasive sampling methods [8].

Our research group has recently developed and validated a non-invasive procedure to identify oral carcinomas from oral brushing samples. Since several diseases have been shown to be associated with differential DNA methylation, including cancers, and epigenetic biomarkers are highly attractive options in clinical practice for their remarkable stability if compared with RNA biomarkers, we decided to investigate a list of genes previously found to be altered in OSCC. Moreover, DNA methylation is closely linked to environmental influences and often altered in the early phase of carcinogenesis, allowing us to detect also OPMD. We used for this purpose a very robust protocol based on bisulfite next-generation sequencing (NGS) [9]. The bisulfite-NGS approach changes unmethylated cytosines (C) to thymines (T), and the methylation ratio was calculated by counting the number of C to T conversions and quantifying the methylation proportion per base. This was achieved by identifying C-to-T conversions in the aligned reads and dividing the number of Cs by the sum of Ts and Cs for each cytosine in the genome. The mean coverage depth was above 1000× to enrich a precise calculation of the C/T ratio. In a seminal study, we quantitatively measured the DNA methylation level of a panel of 13 genes and developed a choice algorithm that discriminates OSCC patients from healthy individuals [10]. A score that exceeded a threshold value (1.0615547) was indicative of epigenetic changes related to the presence of OSCC. The score calculation was described previously by Morandi et al. [10]. A recent multicenter study validated the procedure in an extensive collection of 220 brushing specimens: 93.6% (103/110) of the brushing specimens collected from patients with OSCC lesions were identified in blindness as positive, and 84.9% (90/106) of the samples from healthy individuals were identified as negative [11]. Subsequently, a 13-gene DNA methylation analysis was applied to the oral brushing samples from high-risk OSCC patients in two subsequent preliminary studies to evaluate the role of our procedure as an indicator of oral cancer risk [12,13]. In the first study, we identified a significant relationship between high-grade dysplasia and positive brushing specimen values collected in patients affected by OL [12]. In the second study, a positive result in an oral brushing specimen collected from regenerating mucosa after resecting OSCC was the most powerful variable related to the appearance of a secondary tumor [13]. In the present report, we describe the potential application of our non-invasive tool based on a 13-gene DNA methylation analysis in oral brushing specimens from a patient affected by OPMD who developed primitive and secondary oral carcinoma during the follow-up period. The aim of the present case description was to evaluate the clinical benefits offered by our minimally invasive genetic procedure in the diagnostic and prognostic work-up of a patient at risk of OSCC development.

## 2. Case Presentation

All clinical investigations have been conducted according to the principles expressed in the Declaration of Helsinki. All information regarding the human material used in this study was managed using anonymous numerical codes. The patient gave informed consent. 

### 2.1. Patient History

A 68-year-old, non-smoking man was referred to the Department of Biomedical and Neuromotor Sciences, Section of Oral Sciences in December 2016 for an asymptomatic white wide lesion in the oral mucosa involving the lingual and vestibular gingiva near dental element #37, the left cheek, and a portion of the soft palate (Figure 1a,b). Medical history revealed a previous diagnosis of diabetes and hypercholesterolemia treated with metformin, atorvastatin, and cardioaspirin. On clinical examination, the white lesion was non-removable and homogenous with well-defined borders. No pain or evidence of ulceration was detected during the initial examination. The patient’s medical history was non-contributory. After the clinical evaluation and after excluding potential etiological causes, a provisional clinical diagnosis of OL was made, and an incisional biopsy was performed to achieve a definitive clinical pathological diagnosis. The histological assessment revealed the presence of acanthosis, hypergranulosis, hyperkeratosis, and non-specific inflammatory cells in the absence of dysplastic characteristics (Figure 1c).

Consequently, the lesion was definitively classified as OL based on the criteria described by Warnakulasuriya et al. [3], and the patient underwent clinical follow-up every 6 months. During a routine follow-up visit in March 2019, a proliferative and dyshomogeneous area was noted in the lesion located in the lingual gingiva near dental element #37 (Figure 2a). The patient also reported discomfort and pain corresponding to element #37. An incisional biopsy and histological assessment revealed the presence of a well-differentiated, micro-invasive OSCC (Figure 2b). Complete surgical resection of the OSCC, together with concomitant extraction of element #37, was performed following standard treatment practice [14]. Final pathological classification revealed a pT1N0 OSCC with a clear margin of resection and a low pattern of invasion (P1 on the basis of Chang et al. classification [15], depth of invasion <4 mm, absence of perineural and vascular invasion).

After surgery, the patient was enrolled in a routine oncological follow-up program, and clinical, instrumental, and radiological examinations were administered according to international National Comprehensive Cancer Network guidelines (Figure 3). During routine follow-up 1 year later, a clinically (Figure 4a) and histologically (Figure 4b) confirmed secondary OSCC tumor developed in the anterior area of the gingiva concerning the index tumor, and the secondary OSCC was surgically resected. The patient is currently free from disease but is still undergoing routine oncological follow-up.

During routine follow-up 1 year later, a clinically (Figure 4a) and histologically (Figure 4b) confirmed secondary OSCC tumor developed in the anterior area of the gingiva concerning the index tumor, and the secondary OSCC was surgically resected. The patient is currently free from disease but is still undergoing routine oncological follow-up.

### 2.2. 13-Gene DNA Methylation Analysis

We used our recently developed non-invasive procedure consisting of an oral-brushing minimally invasive sampling procedure and DNA methylation analysis of a preselected panel of 13 genes in the oral mucosa. Brushing samples were collected as described previously [10,12,13,16] at five different times as shown in Table 1: at the time of OL diagnosis, collected from the surface of the white lesion (December 2016); concomitant with the biopsy, which led to the diagnosis of the index OSCC in the proliferative area with homogeneous dye (April 2019); 6 months after surgical removal of the primitive OSCC in the regenerative area following the primary OSCC resection (October 2019); at the time the secondary tumor appeared, collected from the tumor mass (May 2020); and 6 months after surgical removal of the secondary tumor in the regenerative area following the second OSCC resection (December 2020). Brushing cell collection in the regenerative mucosa after the primary and secondary tumor was performed in the area correspondent the original place of OL: vestibular and lingual gingiva, left cheek, and soft palate. A 13-gene DNA methylation analysis was performed as described previously by Morandi et al. [10]. Briefly, DNA from exfoliated cells was purified using the MasterPure Complete DNA Purification Kit™ (MC85200; Lucigen, Middleton, WI, USA) and treated with sodium bisulfite using the EZ DNA Methylation-Lightning Kit™ (D5031; ZymoResearch, Irvine, CA, USA) according to the manufacturer’s instructions. Quantitative DNA methylation analysis of the following genes was performed by next-generation sequencing: *ZAP70*, *ITGA4*, *KIF1A*, *PARP15*, *EPHX3*, *NTM*, *LRRTM1*, *FLI1*, *MIR193*, *LINC00599*, *MIR296*, *TERT*, and *GP1BB*. Libraries were prepared using the Nextera™ Index Kit and a locus-specific bisulfite amplicon approach [10]. The libraries were loaded onto MiSEQ (15027617; Illumina, San Diego, CA, USA). FASTQ output files were processed for quality control (>Q30) and converted into FASTA format in a Galaxy Project environment [17].

The methylation ratio of each CpG was calculated in parallel using different tools: BSPAT (http://cbc.case.edu/BSPAT/index.jsp accessed on 29 December 2020) [18], BWAmeth in a Galaxy Project environment (Europe) followed by the MethylDackel tool (https://github.com/dpryan79/MethylDackel accessed on 29 December 2020), EPIC-TABSAT [19], and Kismeth [20]. In our previous study [10], the best CpGs identified by ROC analysis were used to generate a choice algorithm based on multiclass linear discriminant analysis. This approach allowed us to correctly identify OSCC at a threshold of 1.0615547, which showed the best sensitivity and specificity values (area under the curve = 0.981). Values exceeding the threshold of 1.0615547 were considered positive. The 13-gene DNA methylation analysis revealed a positive result in specimens collected from primary (score 5.21) and secondary (8.14) OSCC. Positive scores were also calculated for the OL lesion diagnosed 28 months before the development of the primary oral cancer (score 1.61) and in the brushing sample collected from the regenerated clinically healthy area 6 months after resecting the primary tumor and 8 months before the appearance of the secondary tumor (score 1.85). Finally, the brushing sample collected from regenerative oral mucosa 6 months after resecting the secondary OSCC was negative (0.47) (Table 1).

## 3. Discussion

Patients diagnosed with OPMD and/or treated for oral carcinoma are considered at high risk of OSCC insurgence, and a life-long follow-up program consisting of visual and tactile assessments is the best option to detect early insurgence of a malignancy.

This case report shows the limits of the current diagnostic procedures in the identification of patients that will undergo malignant transformation and the potential clinical application of a minimally invasive procedure based on the methylation level of a panel of 13 genes in oral brushing specimens in a patient with a primitive diagnosis of OL who developed two metachronous oral malignant manifestations during the follow-up period. The brushing cell collection, 13-gene DNA methylation analyses, and score calculation in this patient were performed at five different times: in December 2016 before the incisional biopsy to confirm the OL diagnosis, in April 2019 before the incisional biopsy confirming the malignant transformation of OL into OSCC, in October 2019 in the clinically healthy mucosa 6 months after surgical resection of OSCC, in May 2020 in the tumor mass of the second cancer, and finally in the clinically healthy mucosa 6 months after surgical resection of the second cancer. Four out of five brushing specimens showed an altered methylation pattern (detected as a methylation score that exceeded the threshold value of 1.0615547). Co-morbidities and medical therapies of the patient did not seem to be responsible for an altered methylation profile of our 13-gene panel (none of 13 genes analyzed were found to be altered expressed in systemic diseases such as diabetes or familiar hypercholesterolemia [21]).

Our procedure detected high scores in brushing samples collected from two metachronous neoplastic manifestations (5.21 for the primary tumor and 8.14 for the secondary tumor) as confirmation of the diagnostic value of our procedure [10]. A positive score (1.61) was also detected in a brushing specimen collected at OL diagnosis 2 years before the neoplastic transformation. Notably, the clinical and histological OL features (homogeneous lesion without the presence of histological dysplasia) did not suggest a substantial risk of malignant transformation. Furthermore, a positive score (1.81) was calculated in the brushing sample collected from the apparently healthy mucosa 8 months before the insurgence of the secondary cancer. These data are in agreement with previous studies and confirm the identified genetic and epigenetic alterations related to the appearance of a secondary neoplastic manifestation in tumor-adjacent tissue or distant clinically and histologically normal mucosa [13,22,23,24,25], but they are in contrast with clinical and histological characteristics of primary OSCC not suggestive of high-risk of relapse (keratinizing-type squamous cell carcinoma of the verrucous type, T1N0M0 with clear margin of resections, absence of perineural infiltration and vascular infiltration depth of invasion <4 mm). Finally, a negative score was detected in the brushing specimen collected from the regenerative clinically healthy area 6 months after removing the secondary OSCC (0.47), and the patient has experienced no further neoplastic manifestations.

In the present case description, 13-gene DNA methylation analysis of oral brushing specimens had diagnostic and predictive power to help the clinician in the screening and longitudinal monitoring of patients at risk of OSCC transformation. Collecting brushing cells at different times (i.e., every 6 months) during the follow-up period may provide more insight related to the oral cancer risk of a single patient, but further studies with adequate follow-up are necessary to confirm this hypothesis.

A non-invasive or minimally invasive procedure based on sampling collection from oral brushing, mouth rinsing, or saliva and analysis of epigenetic markers was recently proposed as a diagnostic aid to identify patients at risk of developing oral cancer [8]. In particular, different authors analyzed the methylation profile of one or more genes as a diagnostic marker in oral malignancy. The majority of studies proposed a study model that included patients with OSCC and non-OSCC normal controls to analyze the diagnostic value of biomarkers in the identification of oral cancer [10,26,27,28,29,30,31,32]. They reported levels of sensitivity of 62–100% and specificity of 46–96%. However, still today, none of these biomarkers has been implemented in the diagnostic work-up, and to the best of our knowledge, there are only two multicenter studies performed with the aim to validate a non-invasive or minimally epigenetic procedure [11,29]. Furthermore, few studies investigate methylation profile in premalignancy with the aim to investigate the predictive value of methylation markers. In particular, some authors demonstrated a significant relationship between the presence of histological dysplasia and an altered methylation profile of multiple genes [12,33,34,35]. Other authors identified epigenetic markers with predictive power in oral premalignancy. In two different prospective longitudinal studies, Chang et al. and Liu et al. showed that p16 methylation was correlated with malignant transformation of OLs with the presence of epithelial dysplasia [36,37]. Juan et al. recruited a number of 171 patients with normal oral mucosa and potentially malignant disorders (with or without the presence of dysplasia) and showed that in a follow-up period of 50.6 months, hypermethylation of ZNF582 was the only significant and independent predictor of disease progression [38]. Finally, two authors analyzed a panel of epigenetic biomarkers with the aim to identify patients surgically treated for OSCC at high risk of secondary neoplastic manifestation. In particular, Righini et al. collected saliva samples every 2–6 months post diagnosis and treatment and showed that hypermethylation of five genes (*TIMP3*, *ECAD*, *p16*, *MGMT*, *DAPK*, and *RASSF1A*) was associated with relapse in OSCC patients [39]. Rettori et al. collected saliva samples immediately after the last curative treatment and at a follow-up visit 6 months after treatment and also identified *TIMP3* promoter hypermethylation as an independent prognostic marker for local recurrence-free survival in patients with head and neck cancer [40].

In the present report, it was also possible to compare the methylation level of the single most informative CpGs from the panel of 13 genes in the brushing specimens collected at five different times. Indeed, as gene methylation status is a reversible process [41], the analysis of premalignant and malignant mucosa during long-term follow-up may provide useful information related to the oral carcinogenesis process.

Our 13 panel is composed of two miRNA of which *MIR296* was previously described to be altered in bladder [42] and lung cancer [43], *MIR193a* in intrahepatic cholangiocarcinoma [44]; the long non-coding *Linc00599*, aliases *MIR124-1HG*, was found to be aberrantly methylated in hematological malignancies [45]. Among the protein-coding genes, *GP1BB* encodes the receptor for von Willebrand factor and mediates platelet adhesion in the arterial circulation; *ZAP70* encodes a tyrosine kinase normally expressed by natural killer cells and T cells, while *KIF1A* encodes a microtubule-dependent molecular motor protein involved in organelle transport and cell division. *PARP15* transfers ADP-ribose from nicotinamide dinucleotide (NAD) to Glu/Asp residues on the substrate protein; *FLI1* encodes a transcription factor, *NTM* the neurotrimin, *TERT* the telomerase; *EPHX3* encodes a protein that catalyzes the hydrolysis of epoxide-containing fatty acids, *LRRTM1* encodes a leucine-rich repeat transmembrane neuronal protein 1, and finally, *ITGA4* encodes a member of the integrin alpha chain family.

Interestingly, *ZAP70* was hypermethylated in all four positive brushing specimens collected, suggesting that an altered methylation level of this gene represents an early and stable event during oral carcinogenesis. In contrast, *KIF1A, TERT*, and *EPHX3* showed aberrant methylation patterns only in brushing samples related to the primary and secondary OSCC. The results regarding *EPHX3* in this patient are in agreement with Guerrero-Preston et al., who demonstrated that *EPHX3* is hypermethylated in OSCC samples only [46].

Based on the current clinical criteria of Hong et al. [47,48], a second neoplastic lesion appeared at the same site less than 2 years after the index tumor was diagnosed as a local recurrence. However, epigenetic data revealed the presence of different pathways between the primary and secondary tumors: 5 out of 13 genes (*LRRTM3, PARP, NTM, ITGA4*, and *MIR193*) showed hypermethylation only in primary OSCC, *GP1BB* showed hypomethylation only in primary OSCC, whereas *MIR296* resulted hypomethylated only in secondary OSCC (see Table 1 for details). Further investigations are necessary to evaluate the role of epigenetic changes as biomarkers to identify the clonal relationships among multiple oral cancers.

## 4. Conclusions

In this study, we describe the clinical application of 13-gene DNA methylation analysis in oral brushing specimens to manage a patient who developed a premalignant lesion and two subsequent neoplastic lesions in the oral cavity over 4 years. A non-invasive or minimally invasive procedure based on molecular markers may be a good diagnostic aid for clinicians to identify and follow-up patients and lesions at risk of malignant transformation. Despite the observation of a single patient in a short follow-up period does not allow to draw definitive conclusions, a trial based on brushing cells collected at different times during the follow-up period in a consistent number of high-risk OSCC patients (patients surgically treated for OSCC) is ongoing to validate the potential role of our procedure as an indicator of disease before the appearance of clinical signs of oral cancer and to assess the correct time test interval.

## Figures and Tables

**Figure 1 diagnostics-12-00284-f001:**
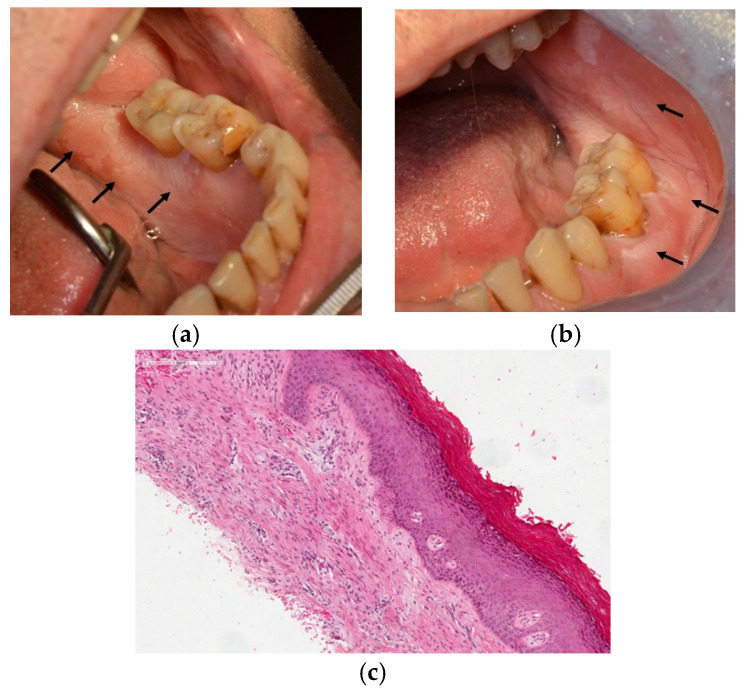
(**a**–**c**) Clinically homogeneous oral leukoplakia (OL): black arrows point out sites of OL extension involving the lingual and vestibular gingiva near dental element #37, the left cheek, and a portion of the soft palate (**a**,**b**). Hematoxylin and eosin staining (HE) of a white lesion showing compact hyperkeratosis and hypergranulosis without dysplasia ( HE 10×) (**c**).

**Figure 2 diagnostics-12-00284-f002:**
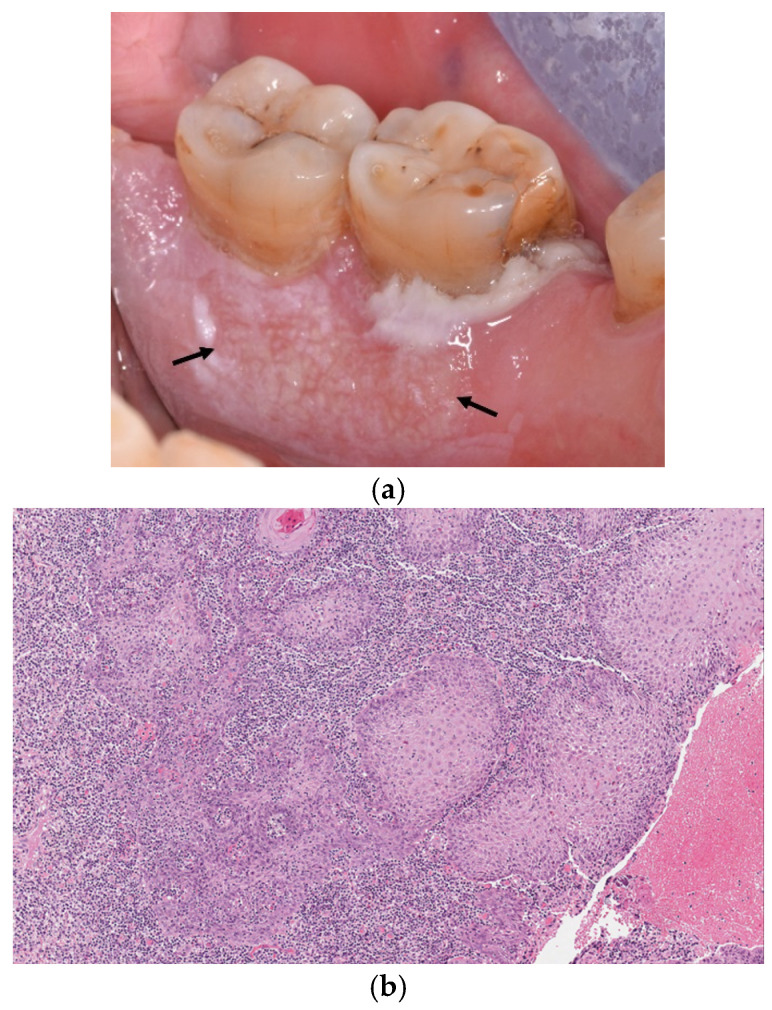
(**a**,**b**) Black arrows indicate non-homogeneous and proliferative lesion involving the lingual gingiva near dental element #37 (**a**). The histological assessment revealed the presence of a well-differentiated, verrucous-type, and keratinizing OSCC with micro-invasive foci (HE 5×) (**b**).

**Figure 3 diagnostics-12-00284-f003:**
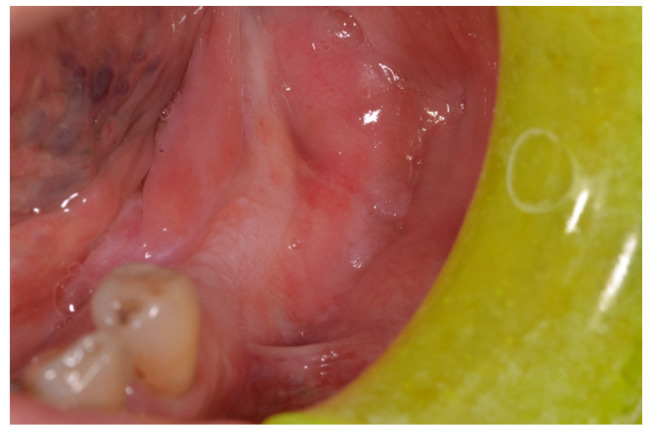
Apparently clinically healthy mucosa 6 months after resecting the OSCC.

**Figure 4 diagnostics-12-00284-f004:**
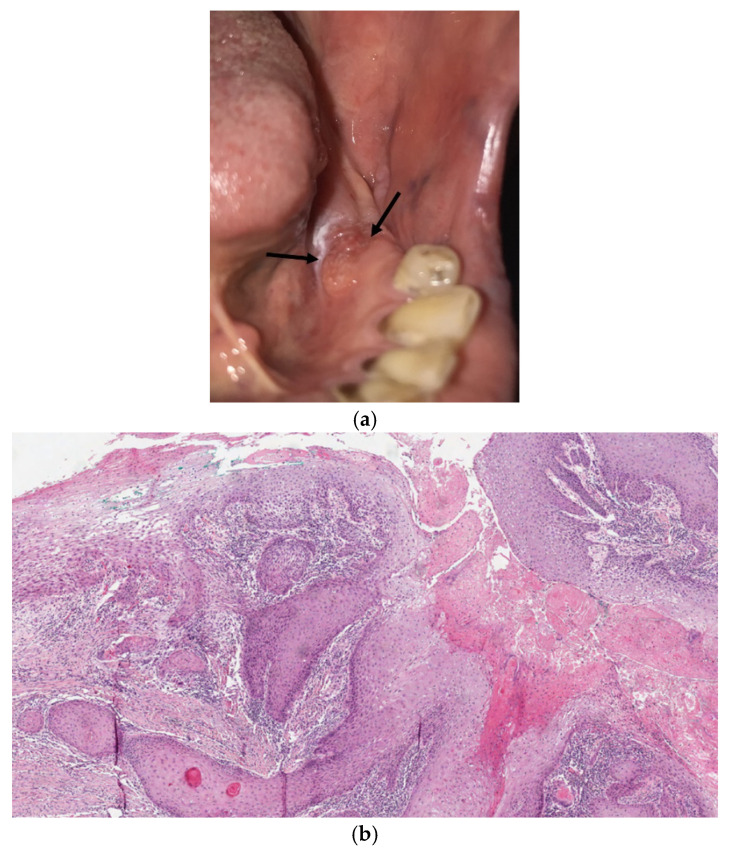
(**a**,**b**) Black arrows showed presence of a proliferative lesion in the area previously surgically treated for OSCC (**a**). The histological analysis revealed the presence of a secondary tumor (HE 5×) (**b**).

**Table 1 diagnostics-12-00284-t001:** Quantitative methylation levels of the most informative CpGs of each gene and methylation scores derived from the algorithm for all the five brushing samples to which the patient was subjected.

Genes	Oral Leukoplakia	Index OSCC	Regenerative Mucosa Six Months after Primary OSCC Resection	Secondary Tumor	Regenerative Mucosa Six Months after Secondary Tumor Resection
Date of brushing sampling collection	December 2016	March 2019	September 2019	March 2020	October 2020
*KIF1A*	0.17647	0.58333	0.0349	0.3942	0.0638
*ZAP70*	0.82342	0.90934	0.998	0.9291	0.7821
*GP1BB*	0.37096	0.79415	0.751	0	0.7586
*LRRTM3*	0.34285	0.33333	0.6538	0	0.1447
*TERT*	0.04323	0.10909	0.0006	0.1897	0
*PARP*	0.03448	0.55445	0.1594	0	0.0054
*FLI1*	0	0	0	0	0
*NTM*	0.15948	0.72847	0.7643	0	0
*LINC0059*	0.03473	0.18089	0.1102	0.1479	0.0015
*EPHX3*	0	0.86813	0	0.6223	0
*ITGA4*	0.59198	0.37885	0.1186	0.0006	0.0405
*MIR193*	0.31818	0.69214	0.5033	0.0699	0.1632
*MIR296*	0.05555	0.04123	0	0.94	0.0005
Methylation score (threshold: 1.061554)	1.61	5.21	1.85	8.14	0.47

## Data Availability

The authors confirm that the data supporting the findings of this study are available within the article and its Appendix A.

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
