# Peer review of "A 13-Gene DNA Methylation Analysis Using Oral Brushing Specimens as an Indicator of Oral Cancer Risk: A Descriptive Case Report"

_diagnostics, 2022, doi:10.3390/diagnostics12020284_

Round 1

Reviewer 1 Report

“A 13-gene DNA methylation analysis using oral brushing specimens as an indicator of oral cancer risk: a descriptive case report”, by Rossi et al is a case study which provides the results of brush biopsy of the oral mucosa of a single patient who progressed from oral leukoplakia to primary OSCC and secondary OSCC. The authors have done nice work in the recent past identifying an OSCC identifying algorithm based on methylation of 13 genes obtained by brush biopsy. They have gone on to show that a subset of lesions show OSCC methylation patterns similar to OSCC suggesting that are at high risk for that disease. This study does resemble a case study in that there is a single patient but unlike other case studies it does not feature an unusal case, a disease that is understudied,  nor does it reveal something new that needs to be noticed right away. Instead it provides an obversation in a single patient that after surgery of the primary tumor the brush biopsy of the surgical field revealed a score associated with OSCC while after the ablation of the secondary tumor, the brush biopsy of the field revealed a score associated with normal tissue. The researchers need to followup on more subjects and then report on that. The new information here is so limited it would not merit a separate report. It would be nice if there were several samples from normal mucosa from this patient to confirm low scores. Tracing tumor evolution is of interest, but considering there is only 1 patient here it would seem they should also do DNA sequencing to look at mutation patterns to see if they concur with what the methylation patterns of the 13 genes show. 

Author Response

Comments and Suggestions for Authors

“A 13-gene DNA methylation analysis using oral brushing specimens as an indicator of oral cancer risk: a descriptive case report”, by Rossi et al is a case study which provides the results of brush biopsy of the oral mucosa of a single patient who progressed from oral leukoplakia to primary OSCC and secondary OSCC. The authors have done nice work in the recent past identifying an OSCC identifying algorithm based on methylation of 13 genes obtained by brush biopsy. They have gone on to show that a subset of lesions show OSCC methylation patterns similar to OSCC suggesting that are at high risk for that disease. This study does resemble a case study in that there is a single patient but unlike other case studies it does not feature an unusal case, a disease that is understudied,  nor does it reveal something new that needs to be noticed right away. Instead it provides an obversation in a single patient that after surgery of the primary tumor the brush biopsy of the surgical field revealed a score associated with OSCC while after the ablation of the secondary tumor, the brush biopsy of the field revealed a score associated with normal tissue. The researchers need to followup on more subjects and then report on that. The new information here is so limited it would not merit a separate report. It would be nice if there were several samples from normal mucosa from this patient to confirm low scores. Tracing tumor evolution is of interest, but considering there is only 1 patient here it would seem they should also do DNA sequencing to look at mutation patterns to see if they concur with what the methylation patterns of the 13 genes show. 

Although the following case presentation doesn’t describe an unusual case or a disease that is understudied it tries to “translate” our previous experimental data from epigenetics into clinical practice, as requested for this special issue collection. Indeed, high-risk oral potentially malignant lesions are generally characterized by the presence of epithelial dysplasia and a non-homogeneous aspect. Instead, the patient described in the present paper showed a flat, homogeneous oral leukoplakia without presence of dysplasia at histological examination. Furthermore, clinical and histological characteristics of the primary OSCC (keratinizing-type squamous cell carcinoma of the verrucous type, pT1N0 with clear margin of resections, low-aggressive pattern of invasion, depth of invasion <4 mm, absence of perineural and vascular invasion) were not suggestive of high risk of relapse. In this context, the identification of an altered methylation pattern based on DNA methylation analysis of our panel of 13 genes (detected as positive score both in oral leukoplakia and in the regenerative oral mucosa after surgical excision of primary OSCC) seems a promising implementation of the traditional clinico-pathological profile for a correct evaluation of the risk of malignant transformation. 

We perfectly agree with the reviewer that the observation of a single patient in a short follow up period is a limitation. For this reason we are currently collecting brushing specimens at different times (every 6 months) from patients surgically treated for OSCC to evaluate the role of our procedure as a reliable measure of oral cancer risk and to assess the correct interval for the test.

In the text we added the following sentences to highlight the limits of the current diagnostic procedures in the identification of patients at risk of malignant transformation

Page 6 line 205.

“This case report shows the limits of the current diagnostic procedures in the identification of the patient that will undergo malignant transformation and the potential clinical application of a minimally invasive procedure based on the methylation level of a panel of 13 genes in oral brushing specimens in a patient with a primitive diagnosis of OL who developed two metachronous oral malignant manifestations during the follow-up period”

Page 7 line 222.

“A positive score (1.61) was also detected in a brushing specimen collected at OL diagnosis 2 years before the neoplastic transformation. Notably, the clinical and histological OL features (homogeneous lesion without the presence of histological dysplasia) did not suggest a substantial risk of malignant transformation. Furthermore, a positive score (1.81) was calculated in the brushing sample collected from the apparently healthy mucosa 8 months before the insurgence of the secondary cancer. This data is in agreement with previous studies that identified genetic and epigenetic alterations related with the appearance of a secondary neoplastic manifestation in tumor-adjacent tissue or distant clinically- and histologically-normal mucosa [13,22–25] but in contrast with clinical and histological characteristics of primary OSCC not suggestive of high-risk of relapse (keratinizing-type squamous cell carcinoma of the verrucous type, T1N0M0 with clear margin of resections, absence of perineural infiltration and vascular infiltration depth of invasion <4 mm).”

 The following paragraph at page 8 line 304 in the conclusions section has been modified to highlight limitations of the present case presentation and future directions.

“A non-invasive or a minimally invasive procedure based on molecular markers may be a good diagnostic aid for clinicians for the identification and follow-up of patients and lesions at risk of malignant transformation, a trial based on brushing cells collected at different times during the follow-up period in a consistent number of  high-risk OSCC patients (surgically treated for OSCC) is ongoing to validate the potential role of our procedure as early predictor of disease (before the appearance of clinical signs) and to assess the correct interval for test repetition.”

Reviewer 2 Report

The authors should add to the patients history information concerning potential medicines taken by the patient and their effect on the results of the study.

Can the authors describe more detailed the statement:  “epigenetic data revealed the presence of different pathways between the primary and secondary tumors"?

Description of statistical analysis should be provided

Author Response

Comments and Suggestions for Authors

The authors should add to the patients history information concerning potential medicines taken by the patient and their effect on the results of the study.

Patient medical history informations have been added at page 3 Line 104

“Medical history revealed a previous diagnosis of diabetes and hypercholesterolemia treated with metformin, atorvastatin and cardioaspirin.”

 A brief sentence related the interaction among systemic diseases, drug therapies and methylation pattern has been added at page 7 line 215

“4 out of 5 brushing specimens showed an altered methylation pattern (detected as a methylation score that exceeded the threshold value of 1.064557). Co-morbidities and medical therapies of the patient don’t seem to be responsible of an altered methylation profile of our 13-gene panel (none of 13 genes analyzed were found to be altered expressed in systemic diseases such as diabetes or familiar hypercholesterolemia).”

Can the authors describe more detailed the statement:  “epigenetic data revealed the presence of different pathways between the primary and secondary tumors"?

The following sentence in the discussion section has been modified at page 8 line 296 for a clearer description of different epigenetic pathways between the primary and secondary tumours:

“5 of 13 genes (LRRTM3, PARP, NTM, ITGA4 and miR193) showed hypermethylation only in primary OSCC, GP1BB showed hypomethylation only in primary OSCC whereas miR296 resulted hypomethylated only in secondary OSCC”

Description of statistical analysis should be provided

Statistical analysis was not performed in the following case presentation. Caption of table 1 has been modified to better clarify the results of DNA methylation analysis of singular CpGs of most informative genes.

Reviewer 3 Report

Introduction: Please elaborate more on oral cancer and OSCC, on the types and influencing factors especially for geriatric patients. 

Please add the gap in the previous literature and findings and how this manuscript will contribute scientifically and in clinics. Also, a section on the aims and objectives of the studies should be included. 

Case preparation: Please add black arrows to point out the regions of interest (ROIs). 

Discussions: Please add the strengths and limitations of this study and future directions. 

Author Response

Comments and Suggestions for Authors

Introduction: Please elaborate more on oral cancer and OSCC, on the types and influencing factors especially for geriatric patients. 

As suggested by the reviewer, the following two sentences have been added in the introduction section:

page 1 line 38

“It usually affects patients in the sixth to eight decades of life, despite recently there is an increasing incidence of OSCC in younger individuals. Elderly OSCC patients generally report usual exposure to well-known risk factors for cancer insurgence such as tobacco or alcohol consumption (PMID: 34759676). Tongue resulted the most common site of OSCC insurgence despite the age of presentation (PMID: 34759676) even if some authors reported alveolar process as predilection OSCC site in elderly patients .

Page 2 line 46

“… and recent studies reported higher recurrence percentages in younger patients compared to elderly ones.” 

Please add the gap in the previous literature and findings and how this manuscript will contribute scientifically and in clinics. Also, a section on the aims and objectives of the studies should be included.

As suggested by the reviewer we added the following sentence to describe the gap and findings in the previous literature at page 7 line 248

“In particular, different authors analyzed methylation profile of one or more genes as diagnostic marker in oral malignancy. Majority of studies proposed a study model that included patients with OSCC and non-OSCC normal controls to analyze the diagnostic value of biomarkers in the identification of oral cancer [11,26–32]. They reported levels of sensitivity of 62-100% and specificity of 46-96%. However still today none of these biomarkers has been implemented in clinical practice and to the best of our knowledge there are only two multicenter studies performed with the aim to validate a non-invasive or minimally epigenetic procedure”

At page 2 line 90 we added the following sentence to better describe how this manuscript will contribute scientifically and in clinics.

“The aim of the present case description was to evaluate the clinical benefits offered by our minimally invasive genetic procedure in the diagnostic and prognostic work up of patient at risk of OSCC development.”

The sentence at page 7 line 252 has been reorganized to better describe the aims and objectives of the studies

“few studies investigate methylation profile in premalignancy with the aim to investigate the predictive value of methylation markers. In particular, some authors demonstrated a significant relationship between presence of histological dysplasia and an altered methylation profile of multiple genes [12,33–35]. Other authors identified epigenetic markers with a predictive power in oral premalignancy: Chang et al. and Liu et al. in two different prospective longitudinal studies showed that p16 methylation was correlated with malignant transformation of OLs with presence of epithelial dysplasia [36,37]. Juan et al. recruited a number of 171 patients with normal oral mucosa and potentially malignant disorders (with or without presence of dysplasia) and showed that in a follow up period of 50.6 months an hypermethylation of ZNF582 was the only significant and independent predictor of disease progression [38]. Finally, two authors analyzed a panel of epigenetic biomarkers with the aim to identify  the prognostic value of epigenetic biomarkers in the identification of patients surgically treated for OSCC at high-risk of secondary neoplastic manifestation…..”

Case preparation: Please add black arrows to point out the regions of interest (ROIs).

Black arrows to point out the regions of interest have been added in figure 1a, 1b, 2a and 4a and relative captions have been modified

Discussions: Please add the strengths and limitations of this study and future directions. 

The paragraph related to Strengths and limitations of this study and future directions has been added in the conclusions section at page 8 line 304

“A non-invasive or a minimally invasive procedure based on molecular markers may be a good diagnostic aid for clinicians to identify and follow-up patients and lesions at risk of malignant transformation. Despite the observation of a single patient in a short follow up period does not allow to draw definitive conclusions, a trial based on brushing cells collected at different times during the follow-up period in a consistent number of  high-risk OSCC patients (patients surgically treated for OSCC) is ongoing to validate the potential role of our procedure as an indicator of disease before the appearance of clinical signs and to assess the correct interval for test repetition.”

Reviewer 4 Report

Thank you for submitting this manuscript. It is an interesting pilot but many additions to the manuscript must be made before publication can be considered. Please find below a list of queries.

  1. A brief explanation on why those particular genes were chosen and their role in the pathway, progression, and diagnosis of OSCC should be given
  2. Explanation about methylation - whether a high number or a low number means increased or decreased gene expression and what is the significance of gene methylation?
  3. The methylation ratio - how were these calculated and what is the scientific principle behind this ratio?
  4. Methylation thresholds and methylation scores- these have to be better explained. Most readers would not be familiar with this concept.
  5. Oral leukoplakia methylation score was even lower than "normal" mucosa at six month post resection. How can this be explained?
  6. Was the regenerative mucosa at six months after primary and six months after secondary resection taken in exactly the same area - and how does this area related to the original place where there was leukoplakia?

Author Response

Comments and Suggestions for Authors

Thank you for submitting this manuscript. It is an interesting pilot but many additions to the manuscript must be made before publication can be considered. Please find below a list of queries.

  1. A brief explanation on why those particular genes were chosen and their role in the pathway, progression, and diagnosis of OSCC should be given
  2. Explanation about methylation - whether a high number or a low number means increased or decreased gene expression and what is the significance of gene methylation?
  3. The methylation ratio - how were these calculated and what is the scientific principle behind this ratio?
  4. Methylation thresholds and methylation scores- these have to be better explained. Most readers would not be familiar with this concept.
  5. Oral leukoplakia methylation score was even lower than "normal" mucosa at six month post resection. How can this be explained?
  6. Was the regenerative mucosa at six months after primary and six months after secondary resection taken in exactly the same area - and how does this area related to the original place where there was leukoplakia?

Actions:

Point 1: the following paragraph was added to discussion page 8 line 275 :

Our 13 panel is composed on two miRNA of which MIR296 was previously described to be altered in bladder (PMID: 22954303) and lung cancer (PMID: 27186308), MIR193a in intrahepatic cholangiocarcinoma (PMID: 29725386); the long non-coding Linc00599, aliases MIR124-1HG, was found to be aberrantly methylated in hematological malignances (PMID: 23406679). Among the protein coding genes, GP1BB encodes the receptor for von Willebrand factor and mediates platelet adhesion in the arterial circulation, ZAP70 encodes a tyrosine kinase normally expressed by natural killer cells and T cells, while KIF1A encodes a microtubule-dependent molecular motor protein involved in organelle transport and cell division. PARP15 transfers ADP-ribose from nicotinamide dinucleotide (NAD) to Glu/Asp residues on the substrate protein; FLI1 encodes a transcription factor, NTM the neurotrimin, TERT the telomerase; EPHX3 encodes a protein that catalyzes the hydrolysis of epoxide-containing fatty acids, LRRTM1 encodes a Leucine-Rich Repeat Transmembrane Neuronal Protein 1, and finally ITGA4 encodes a member of the integrin alpha chain family.

Point 2-3:

The following paragraph was included in the introduction section page 2 line 63:

 Since several diseases have been shown to be associated with differential DNA methylation including cancers, and epigenetic biomarkers are highly attractive options in clinical practice for their remarkable stability if compared with RNA biomarkers, we decided to investigate a list of genes previously found to be altered in OSCC. Moreover, DNA methylation is closely linked to environmental influences and often altered in the early phase of carcinogenesis, allowing us to detect also OPMD. We used for this purpose a very robust protocol based on bisulfite Next Generation Sequencing (NGS) (PMID: 30479381). The bisulfite-NGS approach changes unmethylated cytosines (C) to thymines (T), and the methylation ratio was calculated counting the number of C to T conversions and quantifying the methylation proportion per base. This was done by identifying C-to-T conversions in the aligned reads and dividing number of Cs by the sum of Ts and Cs for each cytosine in the genome. The mean coverage depth was above 1000x to enrich a precise calculation of the C/T ratio.

Point 4:

The score was derived by our developed algorithm.

We added in the introduction section the following sentence page 2 line 77:

The score calculation was described previously by Morandi et al. (Morandi et al 2017 Clin Epigenetics).

Point 5

In all our previous papers we didn’t find a linear relationship between the numeric value of methylation score and an “aggressive behavior” (i.e. the higher methylation score, the greater probability of malignant transformation). However, we demonstrated that methylation scores that exceeded the threshold of 1.0615547 resulted significantly able to discriminate OSCC patients vs healthy donors, OPMDs with presence of dysplasia vs OPMDs without dysplasia and in the follow-up of OSCC treated patients at risk of secondary tumor. Hence, dycothomic characterization of the methylation profile based on the calculated cut off (positive vs negative) so far emerged to be clinically more relevant.

Both methylation scores of oral Leukoplakia (1.61) and regenerative clinically “normal” mucosa (1.85) exceeded the threshold value (1.0615547) and were considered positive. Different studies identified genetic and epigenetic alterations at the surgical margins in formalin-fixed para n-embedded (FFPE) samples and in brushed cells taken from tumor-adjacent or distant clinically- and histologically-normal mucosa. This may explain a high methylation score in regenerative oral mucosa.

A brief sentence to explain the significance of an high methylation value in regenerative normal mucosa has been added at page 7 line 227.

“This data is in agreement with previous studies that identified genetic and epigenetic alterations related with the appearance of a secondary neoplastic manifestation in tumor-adjacent tissue or distant clinically- and histologically-normal mucosa [13,22–25]”

Point 6

Brushing cell collection at six months after primary and secondary OSCC resection was performed in the area correspondent the original place of oral leukoplakia, in particular in the left inferior vestibular and lingual gingiva, in the left cheek and in a part of soft palate.

The following sentence at page 5 Line 169 was added to provide clearer informations on the oral brushing sampling collection for the DNA methylation analysis

“Brushing cell collection in the regenerative mucosa after primary and secondary tumor was performed in the area correspondent the original place of OL: vestibular and lingual gingiva, left cheek and soft palate.”

Round 2

Reviewer 1 Report

A 13-gene DNA methylation analysis using oral brushing specimens as an indicator of oral cancer risk: a descriptive case report" by Rossi et al. is not the standard case report but it is a single case and may be of itnerest. Given that the other reviewers did not have a porblem with them using a single subject and the fact that they have toned down the conclusion I would recommned publishing. 

Reviewer 4 Report

I am happy with additional work included in the manuscript and am happy to accept the paper with minor grammatical changes. 

thank you